# Plastic Interactions with Pollutants and Consequences to Aquatic Ecosystems: What We Know and What We Do Not Know

**DOI:** 10.3390/biom12060798

**Published:** 2022-06-07

**Authors:** Fernanda Cássio, Daniela Batista, Arunava Pradhan

**Affiliations:** 1Centre of Molecular and Environmental Biology (CBMA), Department of Biology, Campus of Gualtar, University of Minho, 4710-057 Braga, Portugal; danimbatista@gmail.com (D.B.); arunava2006molbio@gmail.com (A.P.); 2Institute for Science and Innovation for Bio-Sustainability (IB-S), Campus of Gualtar, University of Minho, 4710-057 Braga, Portugal

**Keywords:** microplastics and nanoplastics, aquatic ecosystems, other contaminants, adsorption, interaction

## Abstract

Plastics are a group of synthetic materials made of organic polymers and some additives with special characteristics. Plastics have become part of our daily life due to their many applications and uses. However, inappropriately managed plastic waste has raised concern regarding their ecotoxicological and human health risks in the long term. Due to the non-biodegradable nature of plastics, their waste may take several thousands of years to partially degrade in natural environments. Plastic fragments/particles can be very minute in size and are mistaken easily for prey or food by aquatic organisms (e.g., invertebrates, fishes). The surface properties of plastic particles, including large surface area, functional groups, surface topography, point zero charge, influence the sorption of various contaminants, including heavy metals, oil spills, PAHs, PCBs and DDT. Despite the fact that the number of studies on the biological effects of plastic particles on biota and humans has been increasing in recent years, studies on mixtures of plastics and other chemical contaminants in the aquatic environment are still limited. This review aims to gather information about the main characteristics of plastic particles that allow different types of contaminants to adsorb on their surfaces, the consequences of this adsorption, and the interactions of plastic particles with aquatic biota. Additionally, some missing links and potential solutions are presented to boost more research on this topic and achieve a holistic view on the effects of micro- and nanoplastics to biological systems in aquatic environments. It is urgent to implement measures to deal with plastic pollution that include improving waste management, monitoring key plastic particles, their hotspots, and developing their assessment techniques, using alternative products, determining concentrations of micro- and nanoplastics and the contaminants in freshwater and marine food-species consumed by humans, applying clean-up and remediation strategies, and biodegradation strategies.

## 1. Introduction

Plastics are a group of synthetic materials made from various organic polymers that can be molded into different shapes (usually by applying pressure and heat) while soft, and they can further set into rigid or slightly elastic forms. Their plasticity and other special properties (from other additives) make plastics useful in a wide range of valuable products [1]. The most common plastic polymers are polyethylene, polyethylene terephthalate, polystyrene, polycarbonate, polypropylene, polyvinyl chloride, polylactide, acrylic, acrylonitrile butadiene, styrene, fiberglass and polyamide (nylon). Plastics have made considerable contribution to global economic growth and have huge societal benefits. The global production of plastics has significantly increased in last five decades and reached 360 million tonnes in 2018, of which 62 million tonnes were produced in Europe, while about 30% of the global plastics were produced by China only [1]. Due to the accelerated production and usage, plastics and plastic particles, in turn, are becoming the key drivers of natural resource depletion, waste, contamination/pollution and are inducing risks and adverse effects to the environment and human health [2].

Plastics consist of H, C, N, Cl and other elements that facilitate their durability and make them non-biodegradable. Because of itsnon-biodegradable nature, the natural degradation of plastic waste may take several thousands of years [3]. Plastics are cost-effective and have become part of our daily life because they are often used in textiles, packaging, automotive, constructions, agriculture, artificial implants, medical appliances, personal care products, etc. and they have various other industrial applications [2,3,4]. In addition, the recent COVID-19 pandemic situation has led to an unprecedented rise in the production and use of face masks and other plastic products [5,6]. To avoid the spreading of this novel coronavirus, various measures, including the use ofsingle-use plastic products, particularly personal protective equipment (PPE), e.g., face masks, face shields, gloves, gowns were implemented worldwide [7]. The improper disposal of these face masks and other COVID-19-related products and medical appliances (made of plastics) are becoming the potential source of microplastic fibers and particles; the single-use face masks, gloves, and gowns are made of various plastic polymers (e.g., polypropylene, polyurethane, polyacrylonitrile, polystyrene, polycarbonate, polyethylene, polyester) [5,7]. The emergence ofthe global production and use of these materials and other plastic products during the pandemichave raised a new challenge in terms ofthe environment and waste management.

In fact, plastics are present in all aquatic compartments including surface water, water column, and sediments, and a significant amount of these are small plastic particles (≤5 mm); thisform of plastics can stay in the environment for a long time [8]. Moreover, the COVID-19 pandemic is likely to act as catalyst for short- and long-term impacts on freshwater ecosystems withthe emergence of face masks, gloves, gowns and other plastic appliances as environmental litter, makingmicroplastic pollution in freshwaters more severe [5,7]. Plastic particles can interact with and adsorb a wide range of pollutants, altering their bioavailability, fate and flux into environmental compartments [9]. Some types of plastic particles have toxic properties in themselves, whereas others contain additives that can leach into the aquatic environment and may induce adverse effects on aquatic biota [9,10,11,12].

The present review aimed to gather information about: (i) the research related to plastic particles and their toxicity in the aquatic environment during the last 10 years in different field categories; (ii) the abundance of plastic particles and their size distribution in freshwaters and seawaters; (iii) the main characteristics of plastic particles that allow different types of contaminants to adsorb on their surfaces;(iv) the interactions of plastic particles with aquatic biota in the absence or presence of different contaminantsand the consequences of their adsorption to plastic particles; and (v) the missing links, challenges and potential solutions which can be helpful in better understanding the current state of knowledge and can boost the research on plastic particles in aquatic ecosystems.

## 2. What Is a Plastic Particle?

Plastics and fibers smaller than 5 mm in size can be considered as plastic particles. The particles with size range between 1 and 5000 μm are defined as microplastics. Plastic particles are generally grouped by usage and origin as (i) primary plastic particles, that are manufactured commercially and utilized as raw materials in the plastic industries or used in abrasives, food packaging and incorporated into other products, e.g., in personal care products; and (ii) secondary plastic particles, that are commonly originated from larger plastic debris, e.g., pieces of cutlery, lids, or various single-use plastic products including face masks [4,13]. Many of these plastic particles are made of polyethylene, but the number of studies monitoring the fragmentation process is limited. Nevertheless, recent studies reported that weathering and some other processes, particularly the exposure of UV radiation in the presence of oxygen, can be the key players of the fragmentation of polyethylene films [7,14]. The deterioration of various plastic litter or discarded plastic products may occur by physical, chemical and biological processes, releasing plastic particles to the environments [2]. The post-fragmented plastic particles are very tiny in size, which means that, after reaching the aquatic environment, they are likely to be mistaken easily as food or as prey by the higher trophic organisms from the food webs, such as fishes; in fact, the majority of the plastics ingested by fishes are found in the form of small particles [15]. Several studies have suggested that, apart from wastewater, surface runoffs, various industrial and hospital effluents and agricultural runoffs, the aeolian transportation and precipitation can also play a key role in accumulating primary and secondary plastic particles into the aquatic environments [2,8,16,17,18].

Microbeads (<1 mm diameter) are considered as primary microplastics that are implemented in various daily use products, such as cosmetics, toothpaste, facial cleansers, and exfoliating soaps [19]. The synthetic polymers used for this kind of plastics are polylactic acid, polyethylene, polyethylene terephthalate, polystyrene and polypropylene. These particles are so tiny that they can easily slip away through water filtering systems [20], causing their release into the aquatic ecosystems, where they can block the digestive tracts of several organisms, including fishes [21], and have the potential to adsorb organic pollutants [22]. Microbeads are ranked the third largest microplastic pollutants (by weight) in the ocean and were found in the Great Lakes, making up about 71% of the total microplastics present [23]. An example of a secondary microplastic is microfiber; microfibers are present in clothes, cigarette butts and diapers, and they also end up into aquatic environments via runoffs or from the synthetic fishing nets, ropes and fishing gears by their abrasive action [24].

Nurdles are commonly known as small plastic pellets (<5 mm); in freshwater ecosystems, the plastic pellets are found as the second-most common type of microplastics [14]. Nurdles are required as a constituent part of almost all plastic products as key elementary units; nurdles are produced by several synthetic polymers (e.g., polyethylene, polypropylene, polystyrene, and polyvinylchloride). Nurdles enter into the environment mostly during shipping, loading and unloading, or transportation, whereby they later reach the water bodies via runoffs. Three decades ago, the Environmental Protection Agency (EPA) warned that nurdles could be deadly products for aquatic organisms, particularly for fishes. Due to their spongy nature, various toxic substances, including PCBs (polychlorinated biphenyls) and DDT (dichloro-diphenyl trichloromethane) can be absorbed by nurdles, causing further impacts on higher trophic organisms (e.g., fish) upon ingestion.

On other hand, foams are lightweight microplastics, composed of mainly polystyrene and commercially applied in packaging materials, foodcontainers, and utensils. Foams also act as carriers of various hydrophilic pollutants/contaminants (e.g., antibiotics, and preservatives) [19]. The chemical substances adsorbed to the surface of these microplastics can be a cause of threat to human health, as those can be released into food, particularly upon applying heat.

Microplastics can be degraded in the environment, which creates nanoplastics (particles ≤ 0.1 µm) by a combination of various methods, including photo-oxidation via exposure to UV rays, high temperatures and increased humidity [25]. Various pro-oxidant transition metals (Mn^2+^, Co^2+^ or Fe^3+^), in the form of stearates, can interact with large plastic debris to form oxo-biodegradable plastics [7]. The incomplete abiotic or biotic degradation of plastics and microplastics may lead to the formation and release the nanoplastics in the aquatic environments [7]. The physicochemical properties (reactivity, conductivity or strength) of nanoplastics are altered upon their formation due to their higher specific surface area. Their decreasing size may constitute a more significant threat to aquatic biota than microplastics; however, due to lack of sufficient available research, as well as efficient detection methods for these particles [26] compared to the larger size plastics, it has become difficult to understand their potential effects on aquatic biota and interactions with other chemical contaminants. Studies have shown that after 45 min of exposure of *Mytilus edulis* and *Crassostrea virginica* to 100 nm size plastics, nanoplastics can stay in their bodies for longer periods compared to microplastics, and can also be translocated to the digestive systems [27].

Despite the significant contributions of plastics to various industries including electronics, healthcare, packaging, construction, automobiles and to human life, inappropriately managed/mismanaged plastic waste has raised a great concern about the ecological risks in aquatic environments over long period and the indirect impacts on human health [3,7,8,28,29] (Figure 1).

## 3. Plastic Pollution in Aquatic Environments

A semi-quantitative analysis based on the search in the Web of Science, using the keywords “Nanoplastics or Microplastics” and “Toxicity and Aquatic” suggests that most research related to plastic particles and their toxicity in aquatic environment has been published in recent years (Figure 2). A clear trend of a hike in the number of studies (publications) has been observed only over the last 4 years (2018–2021). Most of these articles have been published in the categories “environmental science”, followed by “engineering environmental”, “toxicology”, “chemistry multidisciplinary”, “nanoscience nanotechnology”, “water research” and “marine freshwater biology”, indicating the rising concern about micro- and nanoplastics, their physicochemical nature and their fate and impacts on both marine and freshwater ecosystems. However, the number of articles in the category “ecology” is very low (Figure 2), but it is likely to increase in the next few years. Several publications are also coming out in the category “biochemistry molecular biology”, suggesting the increasing research on the interactions of the plastic particles with various organisms at cellular (more specifically macromolecular) levels. The acceleration in the published articles over the last few years highlights the knowledge gap about the long-term ecological consequences of these plastic particles along with other organic contaminants/pollutants in the (aquatic) environment.

According to the United Nations Environment Programme (UNEP), marine litter was defined as “any persistent, manufactured or processed solid material discarded, disposed of or abandoned in the marine and coastal environments”. The UNEP further explained that marine litter consists of objects which were made, used and discarded by humans accidentally (e.g., fishing gear or other plastic materials lost at sea under bad weather) or deliberately into the rivers, beaches, shores and sea/ocean [30]. These objects can also indirectly reach the sea via rivers, runoffs, sewage, storm water or by winds [30].

Plastic litter (debris) in aquatic (coastline, marine and freshwater) environments was considered as a great emerging problem, mainly due to its visibility as flotsam, and the presence in an increasing amount of shores, beaches and the seabed [30]. Plastics have a long half-life and their biodegradation rate is very low due to their polymeric nature and the presence of various plasticizers [31]. The long persistency or durability of plastics in the environment led them to be the materials of choice for various manufacturers. However, plastics can be degraded by prolonged exposure to UV radiation (due to photo-oxidation) and other weathering processes (e.g., high temperature and humidity, decomposition or loss of plasticizers). Nevertheless, the main route of plastic degradation is via mechanical abrasion generating smaller plastic particles (e.g., micro- and nanoplastics). The propensity of these micro- and nanoplastics to have ecological impacts in an aquatic environment has brought them to the forefront of ecotoxicological research [32]. Pollution of plastics and microplastics in surface waters has been reported recently. For example, 1.29 × 10^4^ plastics m^−3^ in the Los Angeles river, San Gabriel river and Coyote Creek, USA, about 8.9 × 10^3^ plastics m^−3^ in Wuhan Lake, China, and about 0.19 plastics m^–3^ in Goiana Estuary, Brazil have been reported, figures which contain a large amount of microplastics [28,33,34]. A huge amount of microplastics has also been found in many European rivers; for instance, ~6, 250 and 350 microplastics m^−3^ have been detected in the Ofanto, Clyde and Henares rivers, respectively [33]. The abundance and size distribution of plastic particles in freshwaters (in the surface waters only) was analyzed based on 30 lines of evidence, of which 15 reported about lake ecosystems [3,35,36,37,38,39,40,41,42,43,44,45,46,47,48], 11 reported about river ecosystems [34,49,50,51,52,53,54,55,56], 2 studied reservoirs [57,58] and 2 studied drinking waters [59,60] (Figure 3A). The abundance and size of plastic particles in surface waters varied widely among different types (even among the same type) of freshwaters; the average distributions of plastic particles were 2.3 × 10^3^ m^−3^ in lakes with the average size of 4.2 mm, 1.27 × 10^5^ m^−3^ in rivers with the average size of 3.4 mm, 2.4 × 10^3^ m^−3^ in reservoirs with the average size of 3.4 mm, and 1.17 × 10^6^ m^−3^ in drinking water with the average size of 0.01 mm. Similarly, also in marine waters (in the surface waters of coastal areas, estuaries, bays, seas, oceans), the abundance and size distribution of plastic particles varied widely (the average abundance was 2.1 × 10^3^ m^−3^ with the average size of 3.7 mm) as analyzed based on 28 lines of evidence [28,61,62,63,64,65,66,67,68,69,70,71,72,73,74,75,76,77,78,79,80,81,82,83,84] (Figure 3B).

Breakdown of plastic litter in aquatic environments may occur naturally via various abiotic and biotic processes. Photodegradation and hydrolysis are the most important abiotic degradation; photodegradation causes oxygenation at the plastic surface and thus it increases its hydrophilicity [85], whereas hydrolysis is promoted by various catalysts or ions (particularly in the wastewater treatment plants or industrial effluents) and, in this process, the water reacts with the plastic polymers inducing physicochemical alteration [85]. Abiotic degradation facilitates the formation of biofilm by microbial communities on the surface of the polymers, which further enhances the biotic degradation [85,86]. Indeed, biotic degradation of polyethylene, polypropylene and polyethylene terephthalate microparticles has been shown in several studies, although the processes are very slow [85,86].

## 4. Characteristics That Facilitate Plastics to Adsorb Different Types of Contaminants

Sorption (specifically adsorption) is the gathering and attachment/adherence of any substance (molecular species) to a solid surface [87]. The surface topology and surface properties of plastics, such as point zero charge, large surface area, acid-base characteristics and functional groups are the key influencing factors for the adsorption of various substances, such as heavy metals, crude oil, sunflower oil, oil spills in marine environments, and dyes [88]. This makes plastic particles as a mean of environmental remediation [89,90] low cost, but also they can act as carriers of contaminants/pollutants (e.g., poly-flouoroalkyl substance, PCBs) putting aquatic life and human health at risk (e.g., Gola et al. [29]).

Moreover, micro- and nanoplastics are considered to impose potential risks to human and environmental health. Due to their tiny size, these plastic particles are becoming ubiquitous in aquatic environments, predicting their bioavailability, direct and indirect entry to the food webs, and accelerated concentrations of harmful contaminants/pollutants by sorption [29]. Sorption of various hydrophobic/non-polar organic substances on micro- and nanoplastics facilitates the chance of these small particles being transported into the aquatic environment and can induce negative effects on aquatic biota and complex ecological processes. However, the sorption of various substances tomicro- or nanoplastic polymers varies with their shapes, types/chemical composition, surface area, surface roughness, size, crystallinity, and binding energy [13,91,92]. Moreover, the sorption of organic and inorganic substances (contaminants/pollutants) to micro- and nanoplastics depend on the physicochemical factors of the aquatic environment (e.g., temperature, pH, hydrophobicity, salinity, diffusivity, and ionic strength [91,93]). The floating micro- and nanoplastic debris are also likely to act as a substrate and vector for various invasive and exotic aquatic species, including microorganisms [31,94], as well as various organic and inorganic contaminants/pollutants [26,95], contaminating otherwise pristine aquatic ecosystems. These chemical contaminants/pollutants can enter the aquatic ecosystems from various sources (e.g., industrial and hospital effluents, wastewater plants, agricultural runoffs, domestic sewage); then, these chemicals can be influenced by various abiotic factors (e.g., chemical oxidation, photolysis, sorption to inert surfaces, hydrolysis) and also by various biotic factors (e.g., sorption to biosurfaces, ingestion by organisms, biodegradation) which, in turn, may decide their fate in aquatic ecosystems. Because micro- and nanoplastics can act as vectors of contaminants, they promote bioaccumulation and bioamplification via the food chain [96].

Many studies have reported that several toxic substances, such as dichlorodiphenyltrichloroethan (DDT), polycyclic aromatic hydrocarbons (PAHs), polychlorinated biphenyls (PCBs), agrochemicals, pharmaceuticals, heavy metals and other persistent organic pollutants adhered to the plastic particles during co-exposure or when collected and removed from freshwater and marine ecosystems [97,98,99,100,101,102,103,104]. The sorption of these toxic contaminants/pollutants from freshwater or marine water to the surface of the plastic particles probably occurred due to the low polarity attraction. Monitoring the sorption mechanisms of specific substances (harmful contaminants/pollutants) under controlled environmental conditions can be useful for better understanding the bioavailability, mobility/transportation, fate and potential risks associated with the substances adsorbed to plastic particles in aquatic ecosystems.

## 5. Interaction of Plastics with Aquatic Biota

Most common examples of the interactions between plastic fragments and aquatic organisms are reported from marine animals (e.g., sharks and turtles) that ingested the fragments of various marine plastic litters (e.g., discarded/accidentally lost fishing gear, and cargo) [30]. Additionally, various riverine- and seabirds pick the plastic debris from water bodies and use those to make their nests; this is another common example of interaction between plastics and animals. Coexposure of aquatic biota to plastic particles and other pollutants/contaminants in waterbodies may lead to toxicological interactions with different magnitude of the adverse impacts [10,105].

The interaction of plastic with aquatic organisms is well recognized despite the limited knowledge on biological impacts of mixtures of plastic particles and other contaminants/pollutants in aquatic environments. To better understand the interactive role of plastic particles as vectors of various contaminants/pollutants to aquatic biota, studies should consider three probable scenarios: (i) contaminated biota consuming clean microplastic; (ii) non-contaminated biota consuming plastics contaminated with other substances; and (iii) contaminated biota consuming contaminated plastics [106]. The potential adverse impacts of micro- and nanoplastics on aquatic ecosystems may vary from their direct effects (i.e., ingestion) due to their capability of adhering and carrying a range of environmental contaminants/pollutants (e.g., pharmaceuticals, agrochemicals, organic and inorganic compounds, and metals).

### 5.1. Impacts of Plastic Particles on Aquatic Biota

Natural biota in aquatic environments may ingest macroplastics and plastic particles by mistake, confusing them with food, many of which are linked to the humans via food chains. For instance, turtles are quite often known to ingest the plastic bags floating on the water, mistaking those for medusa, one of their key food sources. Micro- and nanoplastics have been reported to be ingested by various organisms, such as plankton [107], the mussel *Mytilus edulis* [108], turtles [109], and fish species [21], and sea birds [110]. The ingestion of micro- and nanoplasticmay raise concern for the exposed organisms since these particles can block the feeding appendages or can be accumulated in the gut or digestive tracts, inducing nutritional impairment and adverse physiological impacts. In fact, polystyrene microplastics were reported to induce mortality of shredders and decrease leaf litter decomposition in streams [111] (Table 1).

Nanoplastics are also known to induce toxic effects on aquatic organisms (Table 1). Recent studies have reported size- and concentration-dependent negative effects of different sized (100 nm and 1000 nm) polystyrene nanoplastics on stream fungal communities and associated leaf litter decomposition [117,118]. In mussels (*M. edulis*), a reduction in filtering activity has been reported [113]. In algae, a reduction in CO_2_ and increasing production of ROS (reactive oxygen species) were observed due to the presence of 70 nm polystyrene nanoplastics [113]. Other studies have shown that nanoplastics can cause behavioral, physiological and metabolic changes in fish, such as in the crucian carp (*Carassius carassius*), inhibition of growth in green algae (*Chlorella* sp. and *Scenedesmus* sp.), and decrease in fertility and reproduction rates in fish, in the crustaceans *Daphnia magna* and *Tigriopus japonicas* [114], and also their ingestion by zebrafish larvae causes neurotoxicity and affects their locomotor activity [135]. Microplastics have been demonstrated to induce toxic effects on feeding, fecundity and survival of copepods [11] and to promote neurotoxicity in zebrafish larvae [10].

### 5.2. Effects of Plastic Particles along with Other Contaminants/Pollutants

Ingested micro- and nanoplastics may also expedite the exposure of cells, tissues or organs to various persistent organic pollutants (POPs) and metals [14] that were adsorbed or adhered to the plastic particles or plasticizers leaching from the those particles; this, in turn, may pose a risk to aquatic biota (Table 1). An antagonistic behavior between microplastics (1–5 µm) and Hg was found for several biomarkers in *Corbicula fluminea* (freshwater bivalve), and 6 days of post-exposure recovery in clean medium did not reverse the negative impacts induced by both pollutants [123]. However, synergistic effects between copper and polystyrene (PS) microplastics (0.1 and 20 µm) were observed in zebrafish, where the presence of microplastics and dissolved organic matter (DOM) aggravated Cu toxicity [119]. Synergistic effects were also reported upon co-exposure of early juvenile discus fish to PS microplastics and Cd, as indicated by the activities of antioxidant enzymes [102] (Table 1). Bioaccumulation of Ag in the intestine of zebrafish and rainbow trout was shown after co-exposure to Ag and polyethylene (PE) microplastics (~100 µm) [98,124]. PE microplastics (1–5 µm) and Cu can affect the population growth of microalgae [97]. Recent studies have shown that microplastics can interact with and adsorb nano-silver [12,136], and co-exposure to both these emerging contaminants can affect the leaf litter decomposition, activities of extracellular enzymes, biomass and community structure of aquatic fungi [12]. Most of the studies on interactions of microplastics with other compounds have mainly addressed the POP groups, although data on the sorption of POPs to plastic particles in freshwaters is limited. Exposure to PE microspheres loaded with PCBs did not promote any significant bioaccumulation of these pollutants in the Norway lobster (*Nephrops norvegicus*) even after 3 weeks [126]. However, co-exposure of copepods to PS microplastics and PAHs showed a decrease in toxicity [125]. Interaction between PE microplastics and PAH can delay the PAH-induced mortality on juvenile common goby [127]. However, various recent studies reported that co-exposure of fishes, shell clams, zooplankton and lugworms to PE, low-density PE, PS, or Polyvinyl chloride microplastics and agrochemicals, pharmaceuticals, PCBs, or PAHs can lead to the bioaccumulation of these POPs and can induce oxidative stress (Table 1). Plastic additives are also among the most important groups of chemicals, and the interaction of those with plastic particles may increase their bioavailability. Indeed, Chen et al. [10] demonstrated that the bioaccumulation of bisphenol-A (BPA) was accelerated in zebrafish (particularly in the head and viscera) due to its interaction with nanoplastics. A similar effect was observed upon co-exposure of daphnia to polyamide microplastics and BPA [133] or of rainbow fish to PE microplastics and polybrominated diphenyl ethers (PBDE) [128]. A study with *D. magna* showed that microplastics (4–141 µm) containing diisononylphthalate significantly affected their reproduction and growth [134], despite the effects emerged relatively late in the experiment. Studies on the interactions between nanoplastics and other contaminants are scarce. PS nanoplastics were reported to induce bioaccumulation and toxicity (oxidative damages) of heavy metals (e.g., Cu), pharmaceuticals, PCBs in various fishes and crustaceans at increasing exposure concentrations (Table 1).

The coexistence of micro- or nanoplastics and other chemicals and the interactions between them are of great concern because (i) the susceptibility of the aquatic organisms ingesting these contaminated plastic particles may increase for various diseases, and (ii) some of the chemicals adsorbed to these plastic particles are carcinogenic. Some of these chemicals, including many plasticizers (e.g., bisphenol-A) are either endocrine disruptors or may induce potential toxicity affecting the physiology and reproduction of the organisms [137]. However, little is known about the mechanisms of desorption of these xenobiotics from plastic particles, particularly under complex environmental conditions. These substances may be accumulated and magnified through the food chain, and become highly concentrated in higher trophic levels, particularly in top predators that are often a source of food for humans, posing a potential health problem with socio-economic risks. Figure 4 represents the affinity between different types of plastic particles and other contaminants, e.g., hydrophilic, hydrophobic or heavy metals which can influence their interactions and toxicity to aquatic biota during co-exposure. The representation was adapted from various reports listed in Table 1, and from Bochicchio et al. [138] and Amelia et al. [104].

## 6. Which Are the Consequences of the Adsorption of Chemicals to Plastics?

Micro- and nanoplastics interact with various hydrophobic (non-polar or of low-polarity) organic and inorganic substances in the aquatic environment, including a number of anthropogenic contaminants/pollutants; these substances can be adsorbed and/or accumulated onto the surfaces of macro- and nanoplastics [95,139]. Many of these adsorbed chemicals are likely to attract microbes, particularly bacteria, which can utilize these chemical substances as sources of carbon and energy. In fact, in an aquatic environment, a potential spatio-temporal association between micro- or nanoplastics and some specific kind of bacteria that can metabolize the adhered substances have been observed [140]. Some of these colonized bacterial species, such as *Pseudomonas* sp., are also capable of decomposing microplastics, depending on the type, aging and surface roughness of the particles, although very little is known about the modes of action involved in this biodegradation [95,141]. Among the microbial communities, bacteria are expected to be the key contributors to the process because of their large genetic diversity and capability of evolving novel biodegradation pathways for various xenobiotics over a short period of time.

Dissolved organic matter (DOM) can physically interact with various chemical contaminants/pollutants and is also prone to being adsorbed onto the surface of plastic particles in fresh and marine waters. A major source of marine DOM comes from the exopolysaccharides (EPS) produced and released by phytoplankton (eukaryotic) and bacteria [142]. These EPS can interface with various low polarity chemicals. Additionally, many amino acids, peptides and proteins are associated with these EPS, suggesting their hydrophobic nature; due to this, marine microbial EPS can interact with micro- and nanoplastics and are capable of being adsorbed and accumulated on the surfaces.

In aquatic environment, microorganisms, particularly those in the bacterial community, are among the key initial colonizers of biofilms, not only on biological surfaces but also on submerged non-biological surfaces including those of plastic particles [31,95]. The surfaces of micro- and nanoplastic particles tend to interact with and adsorb hydrophobic substances upon entering into the aquatic environment. On the other hand, biofilm-forming microbial communities prefer non-polar hydrophobic surfaces for adherence compared to the hydrophobic ones [31]. The formation of biofilms on plastic particles is subjected to a succession of phases. The adherence of some submerged chemical substances on the surfaces of plastic particles can lead to the colonization of a bacterial community, particularly of the species with an affinity for the adsorbed substances [31]. Thus, those chemical substances can trigger the colonization of microbial communities on the surfaces of micro- and nanoplastics; this, in turn, can influence the other colonizers (such as fungi, algae, protozoa, dinoflagellates, and diatoms) to develop biofilm [31]. Released exopolysaccharides during biofilm formation adhere to the surfaces of plastic particles and may also act as shield for the microbial communities, which, in turn, may protect them from harsh environments. In fact, micro- and nanoplastics can become the useful substrates for biofilm formation and the adherence of hazardous chemicals can be facilitated due to the long half-life, smaller size, high specific surface area, hydrophobicity and roughness of the surfaces of plastic particles [31]. Therefore, understanding the complex interactions among micro- or nanoplastics, biota associated with biofilm formation, and various organic or inorganic pollutants/contaminants can be crucial for risk assessment in aquatic environment, particularly in wastewaters due to the coexistence of various chemical contaminants (e.g., metals) and plastic particles [95].

## 7. Missing Links, Challenges and Potential Solutions

Because out of 61.8 million tonnes of plastics produced in EU in 2018 (17% of global production), about 29.1 million tonnes of plastic waste were collected annually, of which only about 9.4 million tonnes were recycled inside and outside the EU [143,144], the gap and challenges to deal with the use and the end-of-life phases of plastics are huge [145]. These data also suggest an enormous amount of plastics accumulated in landfills and natural environments [146] (see Figure 2). Handling these steps should include the assessment of the lifecycle of plastic products, monitoring the release of micro- and nanoplastics or pellet losses, and the leaching of chemicals and their ecological impacts [145,147,148].

One of the major reasons for plastic pollution is the poor end-of-life waste management; mismanagement was estimated to occur for 10% of the global plastic waste in 2010, resulting in pollution and missed economic opportunity annually [146,149]. In addition, the major concern about the plastic pollution is the longevity of plastics; most of the plastic litter accumulated in natural environments can persist over hundreds or thousands of years, during which the larger size plastics can be fragmented into smaller size particles (micro- and nanoplastics) [146] (Figure 2). Despite the biochemical inert nature of plastics, a range of chemicals, introduced as additives (e.g., thermal stabilizers, antimicrobial agents, paints, and plasticizers) during the production to achieve the desired product quality, may modify the physicochemical properties of plastics and enhance their durability under adverse environmental conditions [13,96]. High surface area to volume ratio of plastic particles, their lipophilicity and various environmental factors facilitate the sorption of many organic pollutants/contaminants [13,129]. This leads to the unpredictability about the degree of long-term ecological and economical impacts. Because studies on the mechanism of interaction between plastic particles and contaminants, sorption kinetics of the contaminants on micro- and nanoplastics, fate and transportation of these plastic particles with adsorbed contaminants, and efficiency of their end-of-life treatments are scarce [13,146], there is a long way to go before knowing their impacts on aquatic ecosystems and human health. Additionally, various studies suggest that the impacts of microplastic particle ageing on the sorption capacity of contaminants/pollutants may be dependent on their types, chemical nature, time of exposure, and other environmental factors, and therefore, several changes in their interactions may occur over time, due to which, the ecological impacts may vary [13,104]. In addition, there are different bioaccumulation factors of plastic particle-bound contaminants/pollutants in biota, because the internal environments vary among the organisms [104].

Although they are challenging, several potential contemporary approaches are suggested to address these problems: (i) improving waste management, which can be achieved by better collection of plastic waste, facilitating plastic recycling rates and preventing the chemical leaching [86,145,146]; (ii) monitoring the key plastic particles, showing high sorption rate for other contaminants with long persistency under environmental conditions, should be identified, and their life cycle phases, bioavailability, mobility, fate and impact should be studied properly [86,96]; (iii) use of alternative products, such as bio-based or biodegradable plastics may decrease the negative ecological impacts and may open new opportunities in terms of a circular economy, as a key solution for environment sustainability [145,146,150]; (iv) identifying the hotspots, in which the geographical distribution and abundance of micro- and nanoplastic particles in marine, coastal and inland waters should be assessed, and highly polluted areas should be identified where the levels of other organic contaminants should also be monitored [96]; (v) quantitative and qualitative assessment of collected plastic particles-chemical contaminants, in which the interactions between the micro- and/or nanoplastics and contaminants adsorbed to their surface should be investigated [96,104]; (vi) assessment of ingestion of plastic particles in biota collected from the waterbodies, and the adsorbed chemicals on the plastic surface should be thoroughly assessed [96,104]; (vii) methodological standardization, such as optimizing the protocols for plastic particle sampling, characterization and analytical methods for assessing the interactions of adsorbed contaminants [96]; (viii) feeding and trophic transfer studies, in which interactive effects of plastic particle-chemical contaminants at environmentally relevant scenario (e.g., realistic chemical concentrations and complex environmental matrix) should be assessed to determine the biomagnification and the degree of ecological impacts [96,104]; (ix) determining the concentrations of micro- and nanoplastics and the contaminants in freshwater and marine food-species consumed by humans and their post-cooking chemical nature to better understand the potential impacts for human health [96]; (x) in depth in vitro studies on cell lines of humans and aquatic organisms to better understand the behavior and fate of plastic particles, their endogenous additives and adsorbed contaminants on human health [96]; (xi) application of various clean-up and remediation strategies to prevent the release of small particles from various plastic products to the environmental compartments and to improve the collection and cleaning of the plastic particles along with the adsorbed contaminants (particularly at the hotspots) [104,145,146,151]; and finally (xii) implementing biodegradation strategies, which would be a key step in limiting the interactions among the existing plastic particles and various harmful contaminants in the waterbodies and decreasing the number of micro- and nanoplastics [86].

Other aspects that are particularly relevant are related to the potential synergetic or antagonistic adverse effects of the pollutants/contaminants in the aquatic environment along with the micro- and nanoplastics [96] at environmentally realistic concentrations of the mixtures. These aspects should be investigated in model organisms (e.g., daphnia, zebrafish) and in key ecosystem processes, such as primary production and plant litter decomposition, to better understand the actual toxicity of plastics to aquatic organisms and ecosystems. Moreover, efforts should also be made to collect and integrate information based on advanced technologies such as omics (e.g., genomics, metabolomics and transcriptomics) to better understand the alterations due to the interaction between micro- or nanoplastics and pollutants/contaminants at cellular and molecular levels. This will pave the way to achieving a holistic view to obtain a full picture of the effects of micro- and nanoplastics in biological systems in aquatic environments.

## 8. Conclusions

This review clearly suggests that most of the studies related to plastic particles and their toxicity in aquatic environment have been published in last 4 years, although the number of related publications in the field of ecology is still low. The abundance of microplastics and their size distribution in freshwaters and seawaters vary widely. The knowledge about the concentration and size distribution of nanoplastics in surface water is scarce. The interaction of plastic particles with other environmental contaminants/pollutants (e.g., POPs) depends on their physicochemical properties. The sorption of heavy metals, hydrophobic and hydrophilic compounds can vary with the degree of affinity, as shown in Figure 4. This, in turn, can influence the impacts (such as bioaccumulation of the contaminants, lethal or sublethal toxicity, oxidative stress, etc.) on aquatic biota. Micro- and nanoplastics, upon individual exposure, can induce negative effects on aquatic organisms at different trophic levels. Despite the limited number of available studies, the degree of impacts by nanoplastics on aquatic organisms differs from the microplastics, probably because of the higher surface-to-volume ratio. The co-exposure of aquatic organisms to plastic particles and other contaminants mostly induced bioaccumulation of the contaminants and showed either synergistic or antagonistic effects; however, few studies showed no influence of microplastics on the effects of the co-contaminant. Knowledge on the mechanism of interaction between plastic particles and environmental contaminants (or pollutants), their sorption kinetics, fate, transportation and efficiency of their end-of-life treatments is scarce. An urgent implementation of several measures has been suggested to deal with plastic pollution in aquatic ecosystems that include developing proper waste management, monitoring key plastic particles (including nanoplastics), identifying the hotspots, and developing their assessment techniques, increasing the usage of alternative products, measuring concentrations of plastic particles and other co-contaminants (particularly POPs) in freshwater and marine food-species likely to be consumed by humans, and developing clean-up, remediation and biodegradation strategies.

## Figures and Tables

**Figure 1 biomolecules-12-00798-f001:**
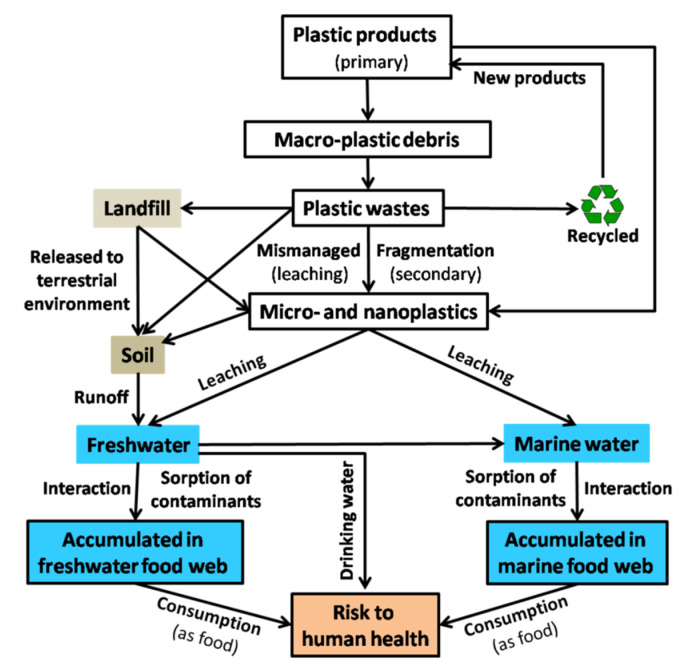
Schematic representation of the formation of plastic particles, their release into aquatic ecosystems and interactions with other contaminants posing risks to environmental and human health.

**Figure 2 biomolecules-12-00798-f002:**
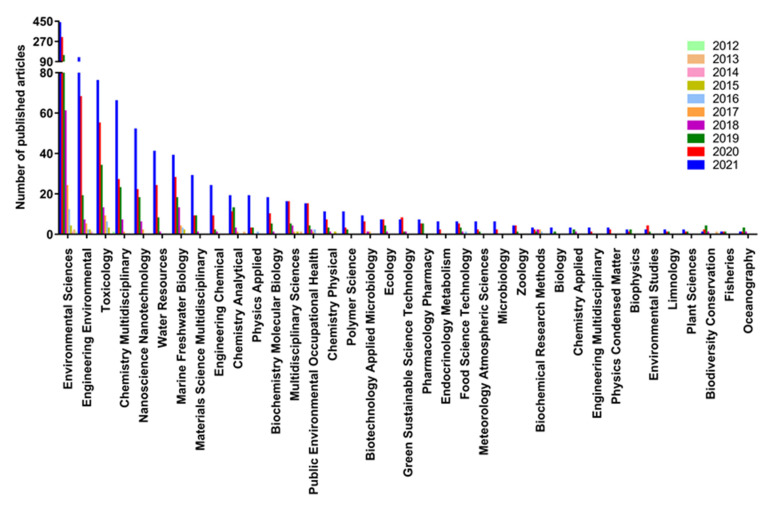
Number of published articles in different Web of Science categories in last 10 years (2012–2021) found using the search keywords “NANOPLASTICS OR MICROPLASTICS AND TOXICITY AND AQUATIC” in all fields. This analysis was performed on 31 January 2022.

**Figure 3 biomolecules-12-00798-f003:**
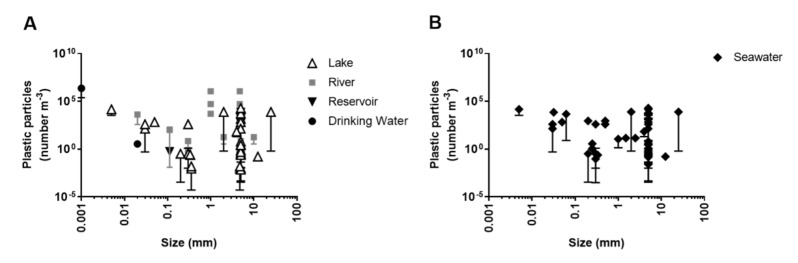
Abundance of plastic particles and their size distribution reported in freshwaters (lakes, rivers, reservoirs and drinking waters) (**A**), and in the marine waters (including coastal areas, estuaries, bays, seas and oceans) (**B**).

**Figure 4 biomolecules-12-00798-f004:**
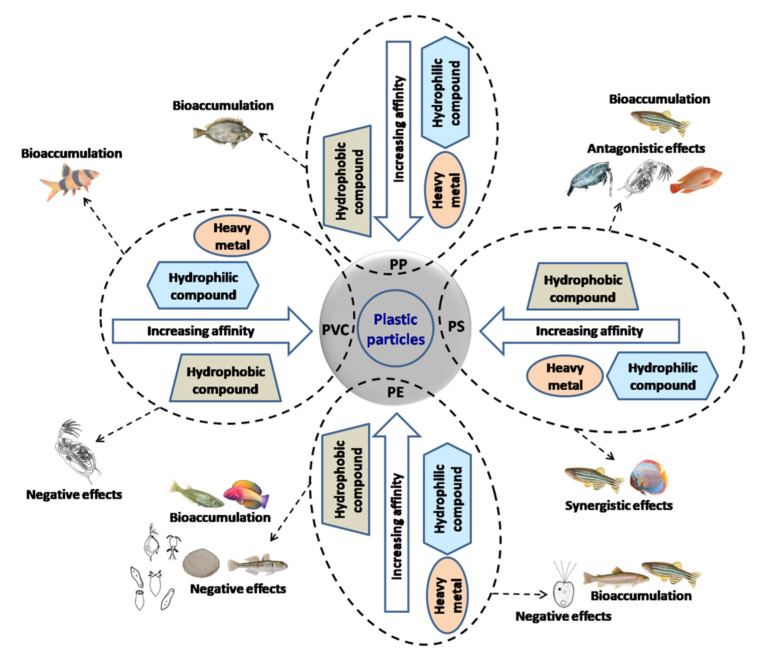
Schematic representation of the affinity between different plastic particles (PP: polypropylene, PE: polyethylene, PS: polystyrene and PVC: polyvinyl chloride) and different contaminants (hydrophilic compounds, hydrophobic compounds and heavy metals) and the impacts of their mixed exposure to aquatic organisms.

**Table 1 biomolecules-12-00798-t001:** Studies assessing the effects of different types of contaminants combined with nano- and micro-plastics. PS—polystyrene, PE—polyethylene, LDPE—low density polyethylene, HDPE—high density polyethylene, PP—polypropylene, PLA—polyamide, PVC—polyvinyl chloride, PBAT—polybutylene adipate terephthalate, PBDE—polybrominated diphenyl ethers, BPA—bisphenol-A, PAHs—polycyclic aromatic hydrocarbons, PCBs—polychlorinated biphenyls, PBDEs—polybrominated biphenyl ethers, PFOS—perfluorooctane sulfonic acid and DiNP—Diisononylphthalate.

Size	Type of Plastic(s)	Contaminant(s)	Organism(s)	Effect(s)	Reference
Nanoplastics	PS 100 nm	none	Blue mussel *(Mytilus edulis)*	Reduction in filtering activity	[112]
	PS 70 nm		Green algae (*Scenedesmus obliquus*)	Reduction of CO_2_ and increasing production of ROS	[113]
	PS 50 and 180 nm		Crustacean (*Daphnia magna*)	High mortality rates and decrease in fertility	[114]
	PS 50 nm		Copepod (*Tigriopus japonicas)*	Decrease in fertility and reproduction rates	[115]
	PS 24 and 28 nm		Crucian carp *(Carassius carassius)*	Behavioral, physiological and metabolic changes	[116]
	PS 100 nm		Aquatic fungal community	Decrease in leaf litter decomposition	[117]
	PS 100 and 1000 nm		Aquatic fungal community	Effects on leaf decomposition and fungal community structure	[118]
	PS 100 nm	Copper (Cu)	Zebrafish (*Danio rerio*)	Synergistic effects (microplastics aggravated Cu toxicity	[119]
	PS 50 nm	BPA	Zebrafish (*Danio rerio*)	Nanoplastics accelerate BPA bioccumulation (head and viscera)	[10]
	PS 100 nm	Pharmaceutical (Roxithromycin)	Red tilapia (*Oreochromis niloticus*)	Nanoplastics increase antibiotic bioaccumulation, affect its metabolism and mitigate neurotoxicity and oxidative damage	[103]
	PS 100 nm	PCBs	Crustacean (*Daphnia magna*)	Lower concentration of nanoplastics decrease toxicity of PCBs, but higher concentration induce lethal effects	[120]
	LDPE < 3 nm	PAHs, PCBs and PBDEs	Japonese medaka (*Oryzias lapides*)	Nanoplastics trigger an increase in bioaccumulation of the contamination	[121]
Microplastics	PS 45 µm	none	Zebrafish larvae (*Danio rerio*)	Neurotoxicity in the locomotor activity	[10]
	PE 10–30 μm and PLA 5–20 μm		Copepod (*T. japonicus*)	Toxic effects on feeding, fecundity, and survival	[11]
	PLA 66 μm and HDPE 103 μm		Oysters (*Ostrea edulis*)	High respiration rates, alteration in benthic assemblage structures	[122]
	PS 10 μm		Caddisfly larvae (*Sericostoma pyrenaicum*)	Mortality of shredders and decrease in leaf litter decomposition	[111]
	Microspheres 1–5 µm	Mercury (Hg)	Feshwater bivalve *(Corbicula fluminea)*	Antagonistic effects in several biomarkers	[123]
	PS 20 µm	Copper (Cu)	Zebrafish (*Danio rerio*)	Synergistic effects (microplastics aggravated Cu toxicity	[119]
	PS 32–40 µm	Cadmium (Cd)	Early juvenile discus fish (*Symphysodon aequifasciatus*)	Synergistic effects (increase in protein carboxyl content, catalase, lysozyme and acid & alkaline phosphatase activities)	[102]
	PE microspheres (19–107 µm)	Silver (Ag)	Zebrafish (*Danio rerio*)	Microplastics decrease uptake of Ag, but induce localization at intestine	[98]
	PE 10–106 µm	Silver (Ag)	rainbow trout (*Oncorhynchus mykiss*)	Ingestion and bioaccumulation of Ag in anterior/mid intestine	[124]
	PE 1–5 µm	Copper (Cu)	Microalgae (*Tetraselmis chuii*)	Decrease in population growth	[97]
	PE < 1 mm extracted from personal care product	Nano-silver (Ag-NPs)	Aquatic fungal community	Decrease in fungal biomass, enzyme activities, leaf litter decomposition	[12]
	PS 10, 50, 100 and 200 µm	PAHs	Copepods (*Acartia tonsa* and *Calanus finmarchicus*)	Microplastics reduce toxicity of PAHs	[125]
	PE, PS microspheres (500–600 or 6 μm)	PCBs	Norway lobster (*Nephrops norvegicus)*	No significant bioaccumulation of the contaminants	[126]
	PE 1–5 µm	PAH (pyrene)	Juvenile common goby (*Pomatoschistus microps*)	Delay in pyrene-induced mortality, reduced acetylcholinesterase and isocitrate dehydrogenase activities	[127]
	PE (size not mentioned)	Agrochemical (paraquat)	Juvenile common carp (*Cyprinus carpio*)	Decrease in total protein, globulin, cholesterol, and triglyceride levels and γ-glutamyl transferase activity	[99]
	PE 10–700 µm	PBDEs	Rainbow fish (*Melanotaenia fluviatilis*)	Induce bioaccumulation of PBDEs	[128]
	LDPE 11–13 µm	PAH (benzo[a]pyrene) and PFOS	Peppery furrow shell clam (*Scrobicularia plana*)	Oxidative damage in digestive gland and gill tissues	[129]
	PE and PP 0.3–5 mm	PCBs and PBDEs	Zooplankton	Induce bioaccumulation	[130]
	PP 0.3–5 mm	PCBs	rabbitfish, (*Siganus rivulatus*)	Induce bioaccumulation of PCBs in muscle tissues, may harm in long run	[131]
	PS 400–1300 µm	PCBs	Lugworm (*Arenicola marina*)	Microplastics affect growth and feeding activity and induce bioaccumulation of PCBs	[132]
	PLA 1–5 µm	BPA	Crustacean *(Daphnia magna)*	Ingestion of contaminants, decreased immobilization	[133]
	PVC 1-10 µm	Pharmaceutical (venlafaxine)	Loaches (*Misgurnus.anguillicaudatus*)	Increase in pharmaceutical bioaccumulation	[101]
	PVC 4–141 µm	DiNP	Crustacean *(Daphnia magna)*	Inhibition on reproduction and growth	[134]

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
