# Peer review of "Plastic Interactions with Pollutants and Consequences to Aquatic Ecosystems: What We Know and What We Do Not Know"

_biomolecules, 2022, doi:10.3390/biom12060798_

Round 1

Reviewer 1 Report

Summary of manuscript biomolecules-1602814 entitled “Plastic interactions with pollutants and consequences to aquatic ecosystems- what we know and what we don´t know”
Summary
In this manuscript, the authors review the field of the impact of micro(nano)plastics on the aquatic environment with special attention to their interaction with other pollutants. The manuscript introduces plastic properties and characterises the main interaction pathways with pollutants. Finally, the authors listed the main challenges related to this research field.
Commentaries
In my opinion, this review is well-written and performs an up-to-date description of the research related to the impact of plastics in the aquatic environment. Therefore, I believe that this manuscript could be eventually published in Biomolecules. First, however, some issues should be corrected by the authors to reflect recent work and improve the clarity of the text.
- I miss more details regarding the interaction mechanisms between plastics and different families of pollutants. For sure, it is the same for metals or POPs? In my opinion, these sections should be enlarged as they are the core of the manuscript.
- Differentiate between the toxicity associated with (macro)plastics, microplastics and nanoplastics. Also, more details should be given regarding the interaction mechanisms of pollutants (organic, metals, …) with a) microplastics, b) nanoplastics. Are they the same? Is there a significant size effect controlling the absorption/adsorption/… Please, discuss.
- Does the degradation of plastics affect these interactions? Discuss.
- It is known that plastics accumulate in some aquatic environments (coastlines, gyres, …). Please, comment on the impact of plastic concentration on these interactions.
- Abstract should be reduced. It is too long and seems like a short introduction to plastics.  

Author Response

Dear Reviewer 1,

Please find our responses below.

Sincerely,

Fernanda Cássio

Summary of manuscript biomolecules-1602814 entitled “Plastic interactions with pollutants and consequences to aquatic ecosystems- what we know and what we don´t know”
Summary
In this manuscript, the authors review the field of the impact of micro(nano)plastics on the aquatic environment with special attention to their interaction with other pollutants. The manuscript introduces plastic properties and characterises the main interaction pathways with pollutants. Finally, the authors listed the main challenges related to this research field.
R: The authors acknowledge the positive comments of the reviewer and appreciate the opportunity to improve the manuscript.

Commentaries
In my opinion, this review is well-written and performs an up-to-date description of the research related to the impact of plastics in the aquatic environment. Therefore, I believe that this manuscript could be eventually published in Biomolecules. First, however, some issues should be corrected by the authors to reflect recent work and improve the clarity of the text.
- I miss more details regarding the interaction mechanisms between plastics and different families of pollutants. For sure, it is the same for metals or POPs? In my opinion, these sections should be enlarged as they are the core of the manuscript.
R: This section has been revised and the aspect highlighted by the reviewer has been addressed in the revised manuscript. However, data on the interactions between micro- or nanoplastics and POPs or other contaminants in aquatic environment and their mechanistic explanation is very limited. A table (Table 1) summarizing the effects of different types of contaminants combined with nano- and microplastics has been included.

- Differentiate between the toxicity associated with (macro)plastics, microplastics and nanoplastics. Also, more details should be given regarding the interaction mechanisms of pollutants (organic, metals, …) with a) microplastics, b) nanoplastics. Are they the same? Is there a significant size effect controlling the absorption/adsorption/… Please, discuss.
R: Unfortunately, comparison among the toxicity associated with (macro)plastics, microplastics and nanoplastics based on interaction mechanism is difficult due to lack of available data on it. Most of the reported studies had either used microplastics or nanoplastics and or used plastic particles with varied type or shape. However, a Table (Table 1) based on earlier studies has been added in which the size, type and effects of micro- and nanoplastics on aquatic biota have been mentioned.      

- Does the degradation of plastics affect these interactions? Discuss.
R: Due to lack of sufficient studies on this aspect, it is difficult to infer about the impacts, and hence this point has not been discussed.

- It is known that plastics accumulate in some aquatic environments (coastlines, gyres, …). Please, comment on the impact of plastic concentration on these interactions.
R: We have included some information on the measured concentrations of plastics and microplastics detected on aquatic environments, particularly on surface water in the revised manuscript.

- Abstract should be reduced. It is too long and seems like a short introduction to plastics.  
R: We followed the reviewer suggestion and the abstract has been revised and shortened

Reviewer 2 Report

This manuscript gives a general overview regarding the plastic interaction with aquatic pollutants as shown in Figure 2. The topic is interesting however the main aim of this review is unknown. The findings also are not clear as they are written neither in the abstract nor conclusion. The review paper in this field should be concise and understandable. I suggest rewriting and resubmission. 

If you want to explain the interaction with heavy metals, oil spills, PAHs, PCBs and DDT, each type should be explained in a different section. 

The interaction of plastic with each type of pollutants should be shown in Table form. What is the specific interaction with different types and compositions of pollutants? After that, what will happen once they attach together? 

Line 15: why only Europe in the abstract?

Lines 15-24: are not recommended to be here. 

Conclusion: Based on the scope of this review you should answer the question in lines 335-338. 

Author Response

Dear Reviewer 2,

Please find our responses below.

Sincerely,

Fernanda Cássio

This manuscript gives a general overview regarding the plastic interaction with aquatic pollutants as shown in Figure 2. The topic is interesting however the main aim of this review is unknown. The findings also are not clear as they are written neither in the abstract nor conclusion. The review paper in this field should be concise and understandable. I suggest rewriting and resubmission. 

R: Thank you for your comments. These points have been taken into consideration and the manuscript has been revised accordingly by reorganizing the abstract and some sections to make the message clearer.

If you want to explain the interaction with heavy metals, oil spills, PAHs, PCBs and DDT, each type should be explained in a different section. 

R: Due lack of sufficient data on these type of interactions and their mechanistic explanation, we added a Table instead where this aspect was considered.

The interaction of plastic with each type of pollutants should be shown in Table form. What is the specific interaction with different types and compositions of pollutants? After that, what will happen once they attach together? 

R: A table has been included showing interactions of micro- and nanoplastics alone or along with different type of pollutants and their effects on aquatic organisms.

Line 15: why only Europe in the abstract?

R: This statement has been removed from the Abstract.

Lines 15-24: are not recommended to be here. 

R: The Abstract has been revised to make it clearer.

Conclusion: Based on the scope of this review you should answer the question in lines 335-338. 

R: The suggestion from the reviewer was not clear to the authors.

Reviewer 3 Report

I have reviewed the manuscript entitled "Plastic interactions with pollutants and consequences to aquatic ecosystems- what we know and what we don´t know" by Cássio et al. for publication in Biomolecules. This review attempts to discuss the current state of knowledge about plastic interactions with pollutants and consequences to aquatic ecosystems. However, the present manuscript offers little that is either new or contributes meaningfully to the current discussion in the literature. The manuscript is overly ambitious in its stated objectives, poorly organized. Although the subject matter is very interesting, there are several issues with the manuscript as currently written that need to be considered:

1)Since this is a standard part of literature search, it may be beneficial to list a concrete and tangible results obtained as a part of this investigation. Authors must explain how this review is unique and different from the others.

2)As a review paper, it requires many figures and/or tables to help support the discussion. The figures included are not of relevance to the topic of the review. Representations of how different pollutants interact with microplastics and the toxicological effects they trigger would be more appropriate.

3)In the various sections there are often repetitions of the same concept without the addition of anything new.

4)In the section 1. “What is a plastic particle?”, there is a very poor introduction to the concept/problem of microplastics. Taking advantage of all the recent literature available, this section should be expanded and improved. Useful literature:

https://doi.org/10.1016/j.envpol.2019.113011; https://doi.org/10.3390/w13070973; https://doi.org/10.1016/j.envpol.2021.116552; https://doi.org/10.1016/j.jhazmat.2020.124357; https://doi.org/10.1021/acs.chemrev.1c00178; https://doi.org/10.1007/s10311-021-01197-9; https://doi.org/10.1016/j.watres.2020.116476; https://doi.org/10.1016/j.jhazmat.2020.124399.

5) Section 2. “Plastic pollution in aquatic environments” is underdeveloped and also not very inherent to the scope of work. It would make sense if it were expanded to make correlations between amount and type of microplates and amount and type of pollutants adsorbed.

6) The sections reporting toxicological studies are very scarce and should be expanded as it is more in line with the title of the paper and the topic of the special issue chosen for publication. Useful literature:

https://doi.org/10.1016/j.fct.2019.111106; https://doi.org/10.3390/toxics9090224; https://doi.org/10.3390/biom11101442.

7)Reference style is incorrect. Moreover, in my opinion, the references are few for a review article.

Author Response

I have reviewed the manuscript entitled "Plastic interactions with pollutants and consequences to aquatic ecosystems- what we know and what we don´t know" by Cássio et al. for publication in Biomolecules. This review attempts to discuss the current state of knowledge about plastic interactions with pollutants and consequences to aquatic ecosystems. However, the present manuscript offers little that is either new or contributes meaningfully to the current discussion in the literature. The manuscript is overly ambitious in its stated objectives, poorly organized. Although the subject matter is very interesting, there are several issues with the manuscript as currently written that need to be considered:

R: Authors appreciate the constructive suggestions and have tried to address the points raised by the reviewer.

1)Since this is a standard part of literature search, it may be beneficial to list a concrete and tangible results obtained as a part of this investigation. Authors must explain how this review is unique and different from the others.

R: We have revised the manuscript by improving most of the sections and included ‘Introduction’ section in which the background and the objectives of this review have been included. In addition, we have also highlighted the major outcomes and the uniqueness of this review in ‘Conclusion’ section of the revised manuscript.

2)As a review paper, it requires many figures and/or tables to help support the discussion. The figures included are not of relevance to the topic of the review. Representations of how different pollutants interact with microplastics and the toxicological effects they trigger would be more appropriate.

R: A schematic representation has been added (see Figure 4) as suggested. Also, the number of figures has been increased and the Table 1 has been updated with additional information in the revised manuscript.

3)In the various sections there are often repetitions of the same concept without the addition of anything new.

R: The sections have been revised adding more information and avoiding repetition.  

4)In the section 1. “What is a plastic particle?”, there is a very poor introduction to the concept/problem of microplastics. Taking advantage of all the recent literature available, this section should be expanded and improved. Useful literature:

https://doi.org/10.1016/j.envpol.2019.113011; https://doi.org/10.3390/w13070973; https://doi.org/10.1016/j.envpol.2021.116552; https://doi.org/10.1016/j.jhazmat.2020.124357; https://doi.org/10.1021/acs.chemrev.1c00178; https://doi.org/10.1007/s10311-021-01197-9; https://doi.org/10.1016/j.watres.2020.116476; https://doi.org/10.1016/j.jhazmat.2020.124399.

R: This section has been revised and some of the suggested articles have been cited to add information. In addition the overview of the formation of plastic particles, their release into aquatic ecosystems, interactions with other contaminants and potential risks to environmental and human health has been shown in Figure 1 under this section in the revised manuscript (formerly it was Figure 2). 

5) Section 2. “Plastic pollution in aquatic environments” is underdeveloped and also not very inherent to the scope of work. It would make sense if it were expanded to make correlations between amount and type of microplates and amount and type of pollutants adsorbed.

R: This section has been further developed in the revised manuscript with additional figures (Figure 2A and B) and text with analyzing the abundance and size distribution of the plastic particles in freshwaters (lakes, rivers, reservoirs and drinking waters) and marine waters (including coastal areas, estuaries, bays, seas and oceans) based on various lines of evidence. This section was intended to focus on gathering the information about the pollution of plastic particles only. Their interactions with other contaminants have been explained in the next sections.

6) The sections reporting toxicological studies are very scarce and should be expanded as it is more in line with the title of the paper and the topic of the special issue chosen for publication. Useful literature:

https://doi.org/10.1016/j.fct.2019.111106; https://doi.org/10.3390/toxics9090224; https://doi.org/10.3390/biom11101442.

R: This section has been expanded adding more information based on the studies listed in the revised Table 1.

7)Reference style is incorrect. Moreover, in my opinion, the references are few for a review article.

R: We tried to be consistent in using the reference style in the revised manuscript.

Round 2

Reviewer 1 Report

Review of manuscript biomolecules-1602814 “Plastic interactions with pollutants and consequences to aquatic ecosystems- what we know and what we don´t know”
I appreciate the effort the authors made to improve the manuscript and to take into consideration the comments and suggestions. 
Therefore, I consider that now the manuscript is suitable for publication. However, the authors should check the numbering on the final table. In addition, the authors must carefully proof-read the text to avoid typos and punctuation errors.

Author Response

Review of manuscript biomolecules-1602814 “Plastic interactions with pollutants and consequences to aquatic ecosystems- what we know and what we don´t know”
I appreciate the effort the authors made to improve the manuscript and to take into consideration the comments and suggestions. 
Therefore, I consider that now the manuscript is suitable for publication. However, the authors should check the numbering on the final table. In addition, the authors must carefully proof-read the text to avoid typos and punctuation errors.

R: The authors appreciate the positive feedback from the reviewer.

Reviewer 2 Report

The authors did not answer my main question. why this review should be published? Unfortunately, I must reject the paper again. 

Conclusion: Based on the scope of this review you should answer the question on lines 392-396. here I mean you should answer your scope and what you found after reviewing the literature. 

Table 1 has a wide range of published papers ( 2012-2022) selected by the authors randomly. It doesn't show the real interaction as most of the studies don't have any contaminants (as you can see in the contaminant column).

What Figure 2 shows? This is another mistake by the authors.

Section 3: lines 247-257. Many studies and cited paper is only one? Lines 249-257 need proper references.

Section 3 title is wrong and should be modified. 

Besides, there are many mistakes in the manuscript that I can not mention one by one. One of them is this reference: Trabulo J, Pradhan A, Pascoal C, Cássio F. 2022. Can Microplastics from Personal Care Products Affect Stream Microbial Decomposers in the Presence of Silver Nanoparticles? Science of the Total Environment (accepted for publication on 31/03/2022). They cited the paper which recently accepted and the authors don't know how to cite it. This paper is online since 4 April.

Author Response

The authors did not answer my main question. why this review should be published? Unfortunately, I must reject the paper again. 

R: We have revised the manuscript by improving most of the sections and included ‘Introduction’ section in which the background and the objectives of this review have been mentioned. In addition, we have also highlighted the major outcomes of this review in ‘Conclusion’ section of the revised manuscript. 

Conclusion: Based on the scope of this review you should answer the question on lines 392-396. here I mean you should answer your scope and what you found after reviewing the literature. 

R: A ‘Conclusion’ section has been added in the revised manuscript highlighting the major outcomes.

Table 1 has a wide range of published papers ( 2012-2022) selected by the authors randomly. It doesn't show the real interaction as most of the studies don't have any contaminants (as you can see in the contaminant column).

R: Table 1 has been revised with additional useful information related to co-exposure and effects of micro- or nanoplastics and other contaminants on aquatic organisms. The text has also been revised by discussing the effects found in these studies.

What Figure 2 shows? This is another mistake by the authors.

R: This figure has been moved as Figure 1 under section 2 (where it fits better and is more justified) in the revised manuscript.

Section 3: lines 247-257. Many studies and cited paper is only one? Lines 249-257 need proper references.

R: This part has been modified and several citations have been included.

Section 3 title is wrong and should be modified. 

R: Done.

Besides, there are many mistakes in the manuscript that I can not mention one by one. One of them is this reference: Trabulo J, Pradhan A, Pascoal C, Cássio F. 2022. Can Microplastics from Personal Care Products Affect Stream Microbial Decomposers in the Presence of Silver Nanoparticles? Science of the Total Environment (accepted for publication on 31/03/2022). They cited the paper which recently accepted and the authors don't know how to cite it. This paper is online since 4 April.

R: This reference has been updated in the reference list of the revised manuscript. When the revised (1st round) manuscript was prepared for submission, it was not available online.

Reviewer 3 Report

Although the authors have made some changes to the manuscript, it still appears to have several shortcomings that make it unsuitable for publication. It is unclear what new aspect the authors have reported on a topic that is so much discussed in the literature. Consequently, I believe that the manuscript should be rejected.  

Round 3

Reviewer 2 Report

The revised version is acceptable. Just the font of Figure 1 should be changed to make it clearer. .